# Human Papillomavirus-Induced Chromosomal Instability and Aneuploidy in Squamous Cell Cancers

**DOI:** 10.3390/v16040501

**Published:** 2024-03-25

**Authors:** Samyukta Mallick, Yeseo Choi, Alison M. Taylor, Pippa F. Cosper

**Affiliations:** 1Department of Pathology and Cell Biology at the Herbert Irving Comprehensive Cancer Center, Columbia University, New York, NY 10032, USA; 2Integrated Program in Cellular, Molecular, and Biomedical Studies, Columbia University, New York, NY 10032, USA; 3Department of Human Oncology, University of Wisconsin School of Medicine and Public Health, Madison, WI 53705, USA; 4Cancer Biology Graduate Program, University of Wisconsin-Madison, Madison, WI 53705, USA; 5Carbone Cancer Center, University of Wisconsin, Madison, WI 53705, USA

**Keywords:** human papillomavirus, chromosomal instability, aneuploidy, chromosome, squamous cell carcinoma

## Abstract

Chromosomal instability (CIN) and aneuploidy are hallmarks of cancer. CIN is defined as a continuous rate of chromosome missegregation events over the course of multiple cell divisions. CIN causes aneuploidy, a state of abnormal chromosome content differing from a multiple of the haploid. Human papillomavirus (HPV) is a well-known cause of squamous cancers of the oropharynx, cervix, and anus. The HPV E6 and E7 oncogenes have well-known roles in carcinogenesis, but additional genomic events, such as CIN and aneuploidy, are often required for tumor formation. HPV+ squamous cancers have an increased frequency of specific types of CIN, including polar chromosomes. CIN leads to chromosome gains and losses (aneuploidies) specific to HPV+ cancers, which are distinct from HPV− cancers. HPV-specific CIN and aneuploidy may have implications for prognosis and therapeutic response and may provide insight into novel therapeutic vulnerabilities. Here, we review HPV-specific types of CIN and patterns of aneuploidy in squamous cancers, as well as how this impacts patient prognosis and treatment.

## 1. Introduction

Squamous cell carcinomas (SCCs) are a group of highly aggressive malignancies that develop in the squamous epithelial cells that line the aerodigestive and genitourinary tracts [1]. Although there is a diverse range of risk factors for the development of SCCs, they can be categorized into two groups: carcinogenic agents, such as UV exposure, alcohol, tobacco, and airborne pollutants, and viral agents, specifically human papillomavirus (HPV) and Epstein-Barr virus (EBV) infection. HPV causes 5% of cancers worldwide, including cancers of the cervix, anus, oropharynx, vagina, vulva, and penis [2].

HPVs are small, non-enveloped viruses encapsulating a small, circular double-stranded DNA genome [3]. Of the five genera of HPV, the genus alpha papillomavirus primarily infects mucosal epithelia and the genera beta and gamma papillomaviruses infect cutaneous epithelia [4]. Alpha HPVs can further be categorized into high-risk and low-risk types based on the specific biological consequences of their infection. Low-risk HPV types, including HPV6 and HPV11, induce the formation of benign warts, while high-risk HPV types, particularly HPV16 and HPV18, act as the etiological agents responsible for various cancers [5]. HPV causes approximately 70% of new head and neck SCC (HNSCC) [6] cases, 91% of anal SCC (ASCC) cases [7], and 95–99% of cervical SCC (CESC) [8] cases. Of note, while HPV-related cancers are most often SCCs, approximately 20% of HPV-associated cervical cancers are adenocarcinomas, which can occur after HPV infection of cervical glandular cells [9]. The biology of squamous and adenocarcinomas are different, thus, we will only be discussing HPV-associated squamous cervical carcinomas here. Although there are many types of HPVs, HPV16 and HPV18 account for the majority of HPV infections in ASCC, CESC, and HNSCC [5]. These high-risk HPV strains infect the basal epithelium through microabrasions and the virus initiates a productive life cycle, including establishment and maintenance phases, followed by productive amplification (reviewed in [10,11]). Fortunately, most hosts are able to clear their infection; however, in some cases, the virus can persist in host cells as episomes or can ultimately integrate into the host DNA. Viral genome integration leads to the constitutive expression of viral oncoproteins E6 and E7, largely due to terminated expression of a viral transcriptional repressor E2 [12]. E2 is involved in tethering the viral genome to host chromatin during mitosis together with BRD4 and TopBP1 and, therefore, also may play a role in malignant transformation [13]. The E6 protein from high-risk HPVs causes the degradation of p53, while E7 perturbs the normal cell cycle by targeting Retinoblastoma (Rb) for ubiquitin-mediated degradation, liberating E2F transcription factors and inducing cells to prematurely enter S phase (reviewed in [3]). Thus, there is increased cell proliferation in the presence of reduced apoptosis and escape of cell cycle checkpoints, all of which contribute to immortalization and malignant transformation.

High-risk HPV E6 and E7 can immortalize cells but are not always sufficient for cellular transformation and cannot directly induce carcinogenesis. Additional genetic insults induced by chromosomal instability and aneuploidy, such as alterations in chromosome number or structure, are necessary for malignant progression. Aneuploidy was the first observed genetic alteration in tumors over one hundred years ago [14]. We now know that almost 90% of solid tumors are characterized by high levels of aneuploidy, including both changes in whole chromosomes and chromosome arms [15,16,17,18]. Aneuploidy is a *state* of abnormal chromosome content that differs from a multiple of the haploid, while chromosomal instability (CIN) represents a *rate* of continuous chromosome missegregation. Thus, CIN always causes aneuploidy but aneuploid cells do not necessarily have CIN. Aneuploid cells can be chromosomally stable if they propagate their abnormal chromosome content faithfully to each daughter cell. Aneuploidy can be quantified using bulk DNA sequencing or by karyotyping, but it is much more difficult to quantify CIN as this requires direct observation of errors during mitosis in fixed or live cells or single-cell sequencing with phylogenetic tree reconstruction.

In general, CIN can be caused by a weakened mitotic checkpoint, centrosome amplification, replication stress, telomere crisis, hyperstable kinetochore-microtubule attachments and defective correction of kinetochore-microtubule attachments, and defects in sister chromatid cohesion and DNA catenation, among other mechanisms described in more detail below (reviewed in [19]). Decreased or absent p53 provides a permissive environment for these errors to occur; thus, HPV has equipped itself well for the induction of CIN. Almost 50% of tumors are characterized by ongoing CIN [20]. When initially induced, aneuploidy and CIN have negative effects on cellular fitness (reviewed in [21]). However, in tumors, it is clear that chromosomal changes have a positive effect on fitness and can be beneficial for tumors [22]. CIN gives cells the opportunity to (1) acquire beneficial aneuploidy events (i.e., deletion of chromosome 10q, where the tumor suppressor PTEN is located) [23]; (2) remove detrimental aneuploidy events; and (3) adapt to different tumor conditions. Ongoing CIN promotes tumor evolution and drives intratumoral heterogeneity and phenotypes that can promote disease progression and treatment resistance [24,25,26]. In most cases, aneuploid clones evolve in a non-random manner, suggesting that they are under selection and represent a karyotype that confers increased fitness. Not surprisingly, chromosomes (or arms) containing oncogenes are enriched and chromosomes containing tumor suppressors are generally lost over time [16]. Additionally, aneuploidy occurs in tissue- and tumor-specific patterns [17,27].

Aneuploidy generally correlates with worse prognosis in multiple tumor types, including lung, breast, and multiple myeloma [28,29,30] (and reviewed in [31]). Ploidy (cellular DNA content) is also an independent prognostic marker in patients with early-stage prostate, colorectal, uterine, ovarian, and breast cancers [32], while high levels of CIN promote metastasis [33]. Several studies have reported that HPV16 E6 and E7 can cause various types of mitotic defects, leading to chromosomal instability and aneuploidy. In this review, we will discuss the mechanisms and implications of HPV-induced CIN and the resulting aneuploidies characteristic of HPV-positive malignancies. How HPV-specific CIN and aneuploidies affect tumorigenesis, tumor evolution, treatment response, and disease outcomes will also be discussed.

## 2. HPV Induces Specific Types of CIN

### 2.1. Polar Chromosomes

Chromosomes chronically misaligned in proximity to spindle poles are termed polar chromosomes, and they arise from a failure in chromosome congression at the metaphase plate during mitosis (Figure 1A). Multiple groups have observed polar chromosomes in cells expressing HPV16 E6 and E7 [34] and HPV+ cervical cells, although they have used different terminology to describe the same phenomenon [35,36]. Polar chromosomes have been observed in dysplastic cervical lesions, and their presence has been suggested as a potential diagnostic marker for HPV-associated cervical intraepithelial neoplasia [35,37,38]. Along these lines, the levels of polar chromosomes increased concordantly with the grade of cervical dysplasia and were highest in invasive carcinoma [35]. Burger et al. showed that the extent to which polar chromosomes occur in cervical intraepithelial neoplasia is associated with the type of HPV, with HPV16+ cells having more polar chromosomes than HPV18+ cells [37]. Recently, we showed that this polar chromosome phenotype is also present in HPV+ HNSCC tumors, patient-derived xenografts, and cell lines [10]. Our studies revealed that HPV16 E6 caused E6AP-dependent degradation of the mitotic kinesin centromere-associated protein-E (CENP-E) [10]. This was also true in cervical cancer cell lines. CENP-E is involved in facilitating proper chromosome alignment at the equator during metaphase, thus reducing CENP-E levels causes the formation of polar chromosomes [39,40]. It is unknown whether HPV16 E6-induced CENP-E degradation causes the missegregation of specific chromosomes or whether the process is random. We hypothesize that this specific type of CIN induced by HPV16 E6 contributes to tumorigenesis by promoting tumor evolution and aneuploidy. High-risk E6 mRNA can be spliced into many different truncated transcripts, named E6*, which have been shown to have both pro- and anti-tumorigenic functions (reviewed in [41]). They are known to increase DNA damage by causing an accumulation of reactive oxygen species, which is likely to increase CIN as it may lead to the formation of acentric fragments. However, this has not been formally tested to the best of our knowledge [42].

### 2.2. Centrosome Amplification and Multipolar Spindles

HPV16 has also been shown to induce centrosome amplification [43,44] (Figure 1A). The centrosome is the microtubule organizing center of the cell and is composed of a pair of centrioles surrounded by a protein complex termed the pericentriolar material. A normal cell contains one centrosome in cell cycle phases G0 or G1. This is duplicated in S phase and the cell then enters mitosis with two centrosomes, each of which will form one spindle pole to allow for a normal bipolar division. HPV16 E7 causes abnormal increased centriole synthesis leading to centrosome amplification and the formation of multipolar spindles [43,44], which others found to be due to the upregulation of PLK4 [45]. Interestingly, unintegrated episomal HPV16 genomes can induce centrosome amplification even at low copy numbers, although this effect is small [46]. HPV16 E7 was also found to co-localize with γ-tubulin, a component of the centrosome, although this was not likely to be a mechanism of centrosome amplification [47].

Abnormal centrosomes are frequently observed in pre-invasive cervical carcinoma [48], and centrosome amplification increases with cervical intraepithelial neoplasia grade during cervical cancer evolution [49]. We and others have observed significant centrosome amplification and multipolar spindles in human HNSCC cell lines and keratinocytes expressing HPV16 E6 and E7 [10,50]. Centrosome amplification has also been observed in anal SCC biopsies from patients, which was associated with increased multipolar spindles and cell division errors [51]. Centrosome amplification-induced formation of multipolar spindles can lead to multipolar divisions, which are associated with significant chromosome missegregation and are often lethal for a cell [52,53]. Alternatively, supernumerary centrosomes can cluster together resulting in two spindle poles, which allows for a bipolar division and increased chance of cell survival. However, this is associated with an increased rate of lagging chromosomes [52]. Thus, supernumerary centrosomes promote chromosome missegregation in many ways, and this is likely a mechanism by which HPV promotes malignant progression.

### 2.3. Chromosome Bridges

Mitotic defects during anaphase include chromosome bridges and lagging chromosomes (Figure 1B). Chromosome bridges can be formed after DNA damage or telomere crisis which, when repaired, often result in dicentric chromosomes. The attachment of each of the two centromeres to opposite spindle poles leads to the stretching of the chromatin between the segregating masses of DNA during anaphase [54]. A failure in the decatenation of two DNA molecules by the enzyme topoisomerase II prior to anaphase can also lead to the formation of chromosome bridges [55]. HPV utilizes the host cell for its own viral replication and thereby hijacks the host DNA damage response. This leads to increased DNA damage and replication stress (reviewed in [56]), both of which cause CIN, including chromosome bridges. Both HPV16 E6 and E7 induce chromosome bridge formation in human keratinocytes, which was thought to be due to the induction of DNA damage [44]. Notably, the co-expression of high-risk E6 and E7 resulted in the highest number of cells with chromosome bridges; however, no significant changes in chromosome bridges were observed in human keratinocytes expressing a low-risk HPV6 E6 and E7 [44]. Additionally, HPV16 E6 and E7 can affect telomerase activity, leading to telomere shortening and the formation of chromosome bridges in human keratinocytes [57]. Interestingly, the E6 protein of cutaneous beta-genus papillomavirus (β-HPVs) can also cause anaphase bridge formation [58], implying that this functionality may be conserved among E6 proteins [10,44].

### 2.4. Lagging Chromosomes and Micronuclei

Lagging chromosomes can occur when a single kinetochore is attached to microtubules emanating from both spindle poles, among other mechanisms. We and others have observed lagging chromosomes in keratinocytes expressing HPV16 E6 [10,44]. To our knowledge, the mechanism of HPV-induced lagging chromosomes is not known. Lagging or otherwise missegregated chromosomes or chromosome fragments can ultimately be enclosed in small, distinct membrane-bound structures called micronuclei [59,60,61] (Figure 1C). Micronuclei formation and frequency have been suggested as a diagnostic marker in pre-cancerous cervical lesions positive for HPV [62,63]. The micronuclei frequencies in cervical epithelial lesions infected with high-risk HPV16, 18, 31, or 33 serotypes were higher than lesions infected with low-risk HPV6 or 11 [62]. Additionally, a positive correlation was observed between the viral load in cervical epithelial lesions and the frequency of micronuclei in patients infected with high-risk HPV [63]. In normal healthy cells, DNA damage is detected by p53, and subsequently, cells are arrested at the G1 phase to inhibit cell cycle progression in the presence of DNA lesions. However, in HPV-infected cells, p53 is degraded by E6, and Rb is inactivated by E7, both of which result in cell cycle progression to S phase even in the presence of DNA damage, increasing the likelihood of CIN and micronuclei formation [64,65,66]. Several serotypes of the β-HPV genus can also induce the formation of micronuclei and subsequently increase tumorigenic mutations in nonmelanoma skin cancer [58]. HPV8 E6 reduces the availability of Bloom syndrome protein (BLM), which is involved in resolving chromosome bridges during anaphase and facilitating faithful chromosome segregation and results in an increased frequency of micronuclei [58,67]. Similarly, HPV38 E6 (also in the β-HPV genus) was shown to be capable of increasing the frequency of micronuclei, although it remains elusive whether the underlying mechanism is identical to that of HPV8 E6 [58]. In summary, numerous studies have shown that micronuclei are a common consequence of CIN elicited by HPV infection. The mechanisms of HPV E6 and E7-induced CIN are summarized in Figure 2.

## 3. Aneuploidy in HPV-Associated Cancer

### 3.1. CIN Leads to Aneuploidy

There are two main types of CIN: numerical and structural. Numerical CIN occurs when whole chromosomes are missegregated via the formation of misaligned, polar, or lagging chromosomes that contain centromeres. This results in a gain or loss of chromosomes in the daughter cell, which results in numerical aneuploidy. Structural CIN is more complex and includes the missegregation of chromosome arms or fragments (centric or acentric), chromosomal translocations, or chromothriptic events. Chromothripsis is a catastrophic DNA damage event that occurs in micronuclei or as a consequence of chromosome bridge formation and leads to extensive genomic rearrangements, which have significant consequences for the cell [68,69,70]. Chromothripsis occurs in most cancer types to different extents, although interestingly, both CESC and HNSCC have a lower frequency of chromothriptic events than many other cancers [71]. The formation of micronuclei can, therefore, lead to both numerical and structural CIN, depending on whether the original missegregation event included a whole chromosome or a fragment. However, once a chromosome or chromatin fragment is enveloped in a micronucleus, it can undergo chromothripsis, thus, numerical CIN can ultimately become structural. Aneuploidy is the ultimate product of CIN; numerical and structural CIN result in numerical and structural aneuploidy, respectively. For example, cells expressing HPV16 developed aneuploidy for chromosome 11 [46], which was due to whole chromosome CIN.

Aneuploidy and copy number alterations (CNAs), defined as changes in the number of chromosome/chromosome arms or any DNA fragments, respectively, are highly prevalent in human cancers, occurring in almost 90% of solid tumors [17]. Chromosomal analyses of several tissue types show that aneuploidy precedes tumor formation [72,73], suggesting that aneuploidy is involved in the early stages of tumorigenesis. In The Cancer Genome Atlas (TCGA) dataset, it has been found that aneuploidy alterations cluster together in tissue-specific patterns, suggesting that specific aneuploidy events have targeted roles in tumor development [17].

### 3.2. Patterns of Aneuploidy

Across the three HPV-associated SCC types, ASCC, CESC, and HNSCC, clear aneuploidy patterns emerge between HPV+ and HPV− tumor samples. As a group, HPV+ tumors have fewer total aneuploidy events compared to HPV− tumors (*p* = 0.03) [74], with an average of seven chromosome arms affected versus twelve. However, HPV+ tumors have a higher fraction of whole chromosome aneuploidies at a median of 33.1% compared to HPV− tumors at 28.7% (*p* = 0.004) [10]. Additionally, certain aneuploidy patterns are frequently observed in SCCs in general, such as chromosome arm 3p, 5q, 8p, 9p, and 18q loss, and chromosome arm 3q and 8q gain [74,75]. While HPV+ SCCs show similar rates of 3q gain, they have significantly lower rates of 3p, 8p, 9p, 17p, and 18q loss, as well as 8q gain [76,77]. HPV+ SCCs also have significantly greater rates of 16q loss than HPV− SCCs [74] (Figure 3).

Rather than randomly occurring in pre-malignant tissues, there are clear patterns and orders in the accumulation of aneuploidy in the development of SCCs [79]. At each stage of progression, distinct chromosome changes are seen. In HNSCC, the order of these genomic alterations differs between HPV+ and HPV− samples [74]. HPV+ tumors show chromosome 1 gain early on in tumorigenesis, whereas 3q, 5p, and 8q gain, and 2q, 3p, 4q, 11q, 17p, and 19p deletion are later events in HPV+ SCCs [80,81,82,83]. Additionally, in CESC, 3q gain is associated with HPV integration into the genome and progression from dysplasia to carcinoma in cervical tissue [84]. It is likely that 9p (containing CDKN2A) loss and 17p (containing TP53) loss are less common in HPV+ tumors due to the fact that HPV E7 mediates Rb inactivation without the need for p16 deletion, and HPV E6 mediates the degradation of p53 without the need for arm-level deletions [3]. On the other hand, in carcinogen-induced HNSCCs, 9p deletion (and *CDKN2A* inactivation) is seen in hyperplastic tissue, whereas 3p and 17p deletion (as well as *TP53* inactivation) then accumulate in dysplastic tissue. In the jump from dysplasia to carcinoma *in situ*, further deletions in 13q, as well as gains in 14q and 11q (along with *CCND1* amplification), are found. As the tumors progress to invasive carcinoma, 4q, 8p, and 10q losses (and *PTEN* inactivation) are seen, as well as 8q gain [79]. Progression from pre-malignancy to malignancy is also associated with 5q, 7q, 18q, 21q, and 22q deletions, as well as 1q, 2q, 3q, 5p, 7p, and 20q gains [85,86,87,88,89].

Aneuploidy is correlated with HPV integration in the genome. A total of 95% of cervical lesions with viral integration were aneuploid, but only 59% of aneuploid lesions had viral integration, suggesting that viral oncogene expression first results in CIN and aneuploidy, followed by HPV integration into the genome of those cells [90]. In cervical lesions, for example, high-risk HPV strains bypass cellular regulatory controls, which allows for E6 and E7 transcriptional deregulation, leading to CIN and eventually aneuploidy. As this process continues, an increasing number of alterations in chromosome number or structure permits the integration of HPV episomal DNA fragments into the host cell genome, which causes viral repressor elements to be lost, leading to the overexpression of HPV oncogenes [12,44,91]. Additionally, upon integration, viral genes and nearby host genes are co-transcribed, leading to increased stability of the transcripts and potential transformative capacity [92,93]. HPV integration does not seem to have a predilection for certain chromosomes, as integration breakpoints have been found across all chromosomes; however, there are certain “hot spots” in the genome where HPV integration occurs more frequently [94,95]. Chromosome 3q happens to be one of the sites enriched for integration events [96,97], and there is some evidence to suggest that there is a slight preference for chromosomal fragile sites throughout the host genome [98]. Interestingly, there is evidence that the extent of HPV integration differs between anatomical sites, as CESC shows higher rates of viral integration than HNSCC [99]. It is not known yet if this can be explained by differences in aneuploidy between these anatomical sites. Overall, HPV infection of squamous cells clearly causes CIN and aneuploidy.

## 4. Therapeutic Implications

While great strides have been made in therapeutic options for SCCs, mortality remains high [100,101]. Early-stage HNSCC, ASCC, and CESC can be treated with surgical resection, which may be followed by a combination of chemotherapy and radiation depending on certain pathological factors. However, locally advanced cases require treatment with definitive chemoradiation. Interestingly, there are clear tissue differences in response to this treatment; most patients with ASCC respond well to radiation, but approximately 30% of patients with CESC do not, as evidenced by persistent disease on post-treatment scans [102]. Moreover, patients with HPV+ HNSCC have a drastically improved response to chemoradiation and have an almost two-fold increase in overall survival compared to patients with HPV-negative disease [103]. In general, HPV-associated cancers tend to have better outcomes than their HPV-negative counterparts in CESC [104] and ASCC [105] as well. A prevailing hypothesis is that HPV-negative cancers often have mutated TP53, while HPV+ cancers may have some remaining wild-type p53 expression [106]. Overall, the reasons for enhanced therapeutic response in HPV+ SCCs are not completely understood [107]. Of note, SCCs in general rarely have actionable oncogenic mutations suitable for targeted therapy [108].

Recent developments in immunotherapy have led to FDA approval of immune checkpoint inhibitors to treat HPV-associated cancers. Immunotherapy has shown some improvement in survival for SCCs, but it is seemingly irrespective of HPV status. In CESC (persistent, recurrent, or metastatic), survival is significantly increased with PD-LI blockade, although the overall response is approximately only 20% [109,110]. Similar results have been observed in HNSCC, although interestingly there is no clear difference in response between HPV+ and HPV− tumors [111,112,113]. It had been hypothesized that HPV+ tumors would respond better to immunotherapy due to a higher neoantigen load [114] and lower overall aneuploidy [115,116,117]. However, HPV can also be immunosuppressive and can modulate interferon responses (reviewed in [3,53]). Perhaps the interaction of all these factors results in no correlation between immunotherapy response and HPV status.

Increased aneuploidy load tends to correlate with a worse prognosis, although this is not always the case. Analysis of dysplastic lesions in the cervix found that lesions with aneuploidy (as measured by flow cytometry) were much more likely to progress compared to diploid lesions [118]. Chromosomal gains are negative prognostic markers for disease-free survival (DFS) and overall survival (OS) in HNSCC (*p* = 0.01 and 0.05, respectively), and 3q gain alone is sufficient for this correlation [74]. However, 16q loss, more common in HPV+ HNSCC, is a significant marker of better DFS and OS (*p* = 0.01 and 0.008 respectively) [74]. In HPV− HNSCC, the deletion of 9p is a biomarker for immunotherapy resistance [119,120,121], attributed at least in part to the CDKN2A/interferon locus at 9p21.3 and the Jak/PD-L1 encoding locus at 9p24.3 [122]. This has been observed in other cancer types as well, including non-small cell lung cancer [123]. Furthermore, CIN has also been associated with immune suppression as well as increased metastasis via chronic activation of the cGAS-STING (cyclic GMP-AMP synthase-stimulator of interferon gene) pathway [33,124].

Beyond HPV-related cancers, the impact of aneuploidy and CIN on therapeutic response is an area of active investigation across all cancer types. Low to moderate levels of CIN are associated with poor prognosis and tumor initiation [125,126]. Conversely, tumor cells with high levels of CIN are more sensitive to ionizing radiation and paclitaxel [127,128,129]. Aneuploidy has been shown to promote resistance to chemotherapy [24,25,26]. Aneuploidy is also inversely correlated with intratumoral immune cell infiltration, and aneuploid tumors tend to have worse responses to immunotherapy [115,117]. An interaction between established immune checkpoint blockade markers, tumor mutation burden (TMB), and aneuploidy is being explored [115]. In general, tumors with low CNA and high TMB tend to respond best to immune checkpoint blockade [130]. Interestingly, highly aneuploid lung tumors are more resistant to radiation alone but have a better response to concurrent radiation and immune checkpoint blockade [116].

The development of therapeutic targets specific for tumors with high CIN or aneuploidy is also an area of active investigation. Since aneuploidy results in proteotoxic stress, proteasome inhibitors have been explored to target highly aneuploidy tumors [131]; unfortunately, these drugs have not been successful in the clinic outside of multiple myeloma [132]. Recently, several groups identified KIF18A, a mitotic kinesin, as a unique dependence and therapeutic vulnerability for whole-genome doubled aneuploid cells [133,134]. Clinical trials using this inhibitor in high CIN cancers are currently ongoing.

## 5. Conclusions

HPV and chromosomal instability have long been known to drive SCC progression. Yet, the interaction between HPV infection and tumor aneuploidy is only beginning to be appreciated. Here, we reviewed the evidence of HPV’s influence on different types of CIN and HPV-specific aneuploidy patterns in SCCs. Initial studies suggest that HPV and aneuploidy together could serve as predictors of therapeutic response. Future mechanistic work may identify clinically actionable vulnerabilities for this patient population.

## Figures and Tables

**Figure 1 viruses-16-00501-f001:**
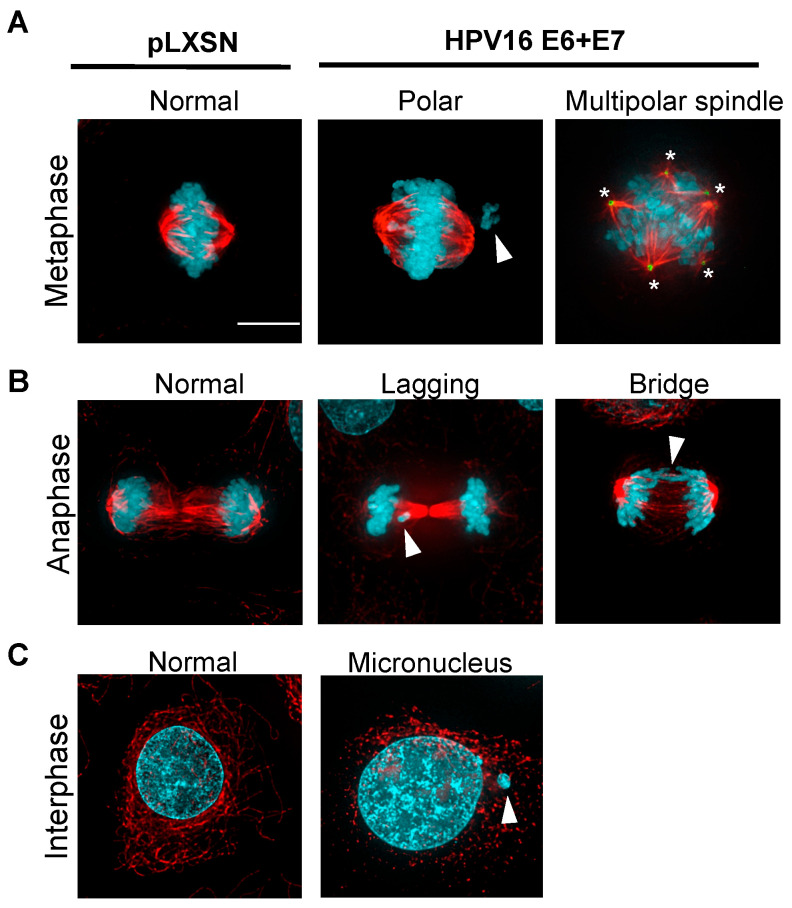
HPV16 E6 and E7 expression in immortalized normal oral keratinocytes (NOKs) induces CIN. (**A**) NOKs expressing the control pLXSN vector (**left**) or HPV16 E6 and E7 (**middle** and **right**) in metaphase. E6 and E7 expression cause polar chromosomes (**middle** panel, arrowhead) and centrosome amplification leading to multipolar spindles (**right** panel, asterisks). Centrosomes at spindle poles are indicated by staining with pericentrin (green). (**B**) NOKs expressing the control pLXSN vector (**left**) or HPV16 E6 and E7 (**middle** and **right**) in anaphase. E6 and E7 expression results in chromosome missegregation, including lagging chromosomes (**middle**; arrowhead) or chromosome bridges (**right**; arrowhead). (**C**) Micronucleus in NOKs expressing HPV16 E6 and E7 in interphase (**right**; arrowhead). NOKs expressing the control pLXSN vector or HPV16 E6 and E7 were fixed with paraformaldehyde, incubated with anti-tubulin or anti-pericentrin antibodies, and counterstained with DAPI as in [10] (blue, DAPI; red, tubulin; green, pericentrin). All images were acquired using a Nikon Eclipse Ti-E inverted fluorescence microscope with a 100×/1.4 numerical aperture oil objective. Images are maximum projections of 0.2 μm z-stacks that have been deconvolved. Scale bar, 10 μm.

**Figure 2 viruses-16-00501-f002:**
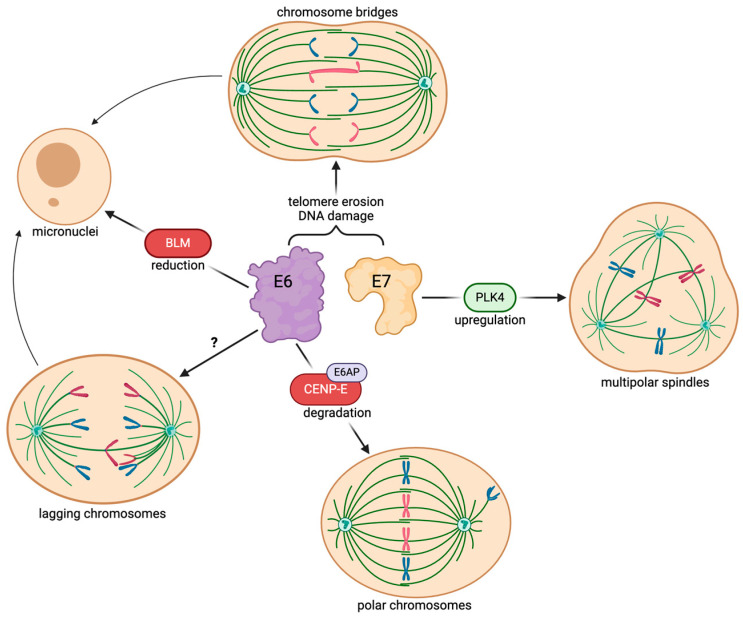
HPV E6 and E7 induce many different types of CIN via unique mechanisms. Created with BioRender.com.

**Figure 3 viruses-16-00501-f003:**
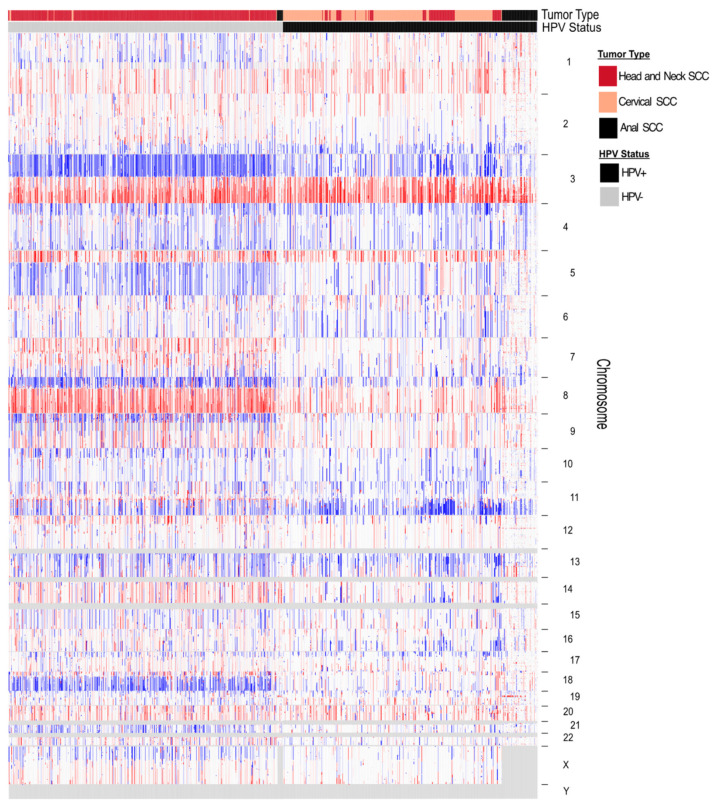
Squamous cell carcinomas show different patterns of aneuploidy depending on HPV status. Integrated Genomics Viewer (IGV) plot of somatic copy number alterations by HPV status for squamous cancer tumor types associated with HPV. Data are sorted by HPV status; tumor type is specified but not clustered. Blue indicates copy number loss, red indicates copy number gain, white indicates copy number neutral, and gray indicates no data available. Sex chromosome copy number data were not available for ASCC. CESC (TCGA, *n* = 242); HNSCC (TCGA, *n* = 501); ASCC (DFCI, *n* = 60). ASCC tumor sample copy number data were made available by Mouw et al. [78]. CESC and HNSCC tumor sample copy number data were made available by The Cancer Genome Atlas [75].

## Data Availability

Head and neck and cervical SCC copy number data are from TCGA, available here: https://gdc.cancer.gov/about-data/publications/pancanatlas (accessed on 4 February 2024). Anal SCC copy number data are from Mouw et al., 2017, available here: https://github.com/vanallenlab/2016-Mouw_ASCC/tree/master/storage (accessed on 4 February 2024).

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
