# Peer review of "Human Papillomavirus-Induced Chromosomal Instability and Aneuploidy in Squamous Cell Cancers"

_viruses, 2024, doi:10.3390/v16040501_

Round 1
Reviewer 1 Report
Comments and Suggestions for Authors
Thank you for the invitation to evaluate the review by Mallick et al. The authors summarize the current status of chromosomal instability and aneuploidy in (squamous) cell carcinomas and in particular under the influence of HPV. They also discuss potential therapeutic options. The review is very well structured and readable. To the best of my knowledge, (almost) all the key points on this topic are addressed. Nevertheless, I would like to mention a few points that the authors might consider including in their text. However, these mechanisms are largely not understood and therefore represent hypotheses for further research.
During malignant transformation from premalignant lesions (also CIN in the cervix for confusion) to tumors, there is a shift in the expression of E6 to its splice variant E6*I. In addition, the expression of E6 or E6*I correlates with the integration status of HPV and, as mentioned in the text, a 3q gain. E6*I is NOT able to degrade p53. On the other hand, E6*I induces cellular stress reactions, which should favor further cell damage including chromosomal alterations.
Campbell et al, (pmid: 29617660) have analyzed TCGA data across squamous cell carcinomas. For the subgroup of HPV-positive SCC, the expression of deltaNTP63alpha (3q!) was also predominant. In addition, HPV genome integration does not appear to be completely random. TP63 and the neighborhood around 3q28 are noticeably frequently affected. Since the alpha variant is explicitly an important differentiation regulator with far-reaching consequences for squamous epithelia, the question arises, which genome-organizing structures could be altered here and favor this genomic region.
In addition, the HrHPV E2 is important for binding to the chromosomes and together with BRD4 there are many regulatory and mechanistic consequences, including DNA repair. The HPV genome is therefore not completely randomly distributed in the nucleus and a loss of the E2 protein due to methylation events or HPV integration and thus E2 gene loss is likely to be relevant in the course of malignant transformation on the genome interaction level.
Reviewer 2 Report
Comments and Suggestions for Authors
The review is well written and summarize the knowledge about chromosomal instability and aneuploidy in SCC cancers induced by HPV.
I would have just one suggestion : to add a figure summarize the effect of HPV E6 and E7 in chromosomal instability and aneuploidy.
Reviewer 3 Report
Comments and Suggestions for Authors
This is a comprehensive and in-depth review of chromosomal instability and aneuploidy in HPV-induced squamous cell cancers. The manuscript is well written and easy to follow. It clearly defines all terms, provides mechanistic insight and points out key outstanding questions. I believe it will serve as a helpful resource for a broad audience interested in SCCs.
A few minor suggestions:
· (page 4) It may be worth pointing out that chromosome bridges can also arise from failure to properly decatenate DNA during mitosis, not exclusively dicentric chromosomes.
· (page 4) I think it makes more sense to pair lagging chromosomes in the micronucleus section rather than with chromosome bridges, since micronuclei, rather than bridges, are a direct consequence of lagging chromosomes.
· (page 5) It may be worth pointing out that chromothripsis can arise from chromosome bridges as well as micronuclei (Maciejowski et al., Cell 2015; Umbreit, et al., Science 2020).
· (page 9) Can also consider mentioning the work by Bakhoum group shows the effect of CIN on cGAS-STING pathway.
